# Escalating progression of mental health disorders during the COVID-19 pandemic: Evidence from a nationwide survey

Li Ping Wong[ORCID][1]*, Haridah Alias[1], Afiqah Alyaa Md Fuzi[2], Intan Sofia Omar[2], Azmawaty Mohamad Nor[3], Maw Pin Tan[4], Diana Lea Baranovich[3], Che Zarrina Saari[5], Sareena Hanim Hamzah[6], Ku Wing Cheong[7], Chiew Hwa Poon[7], Vimala Ramoo[ORCID][8], Chong Chin Che[8], Kyaimon Myint[9], Suria Zainuddin[10], Ivy Chung[2]*

1 Centre for Epidemiology and Evidence-Based Practice, Department of Social and Preventive Medicine, Faculty of Medicine, University of Malaya, Kuala Lumpur, Malaysia, 2 Department of Pharmacology, Faculty of Medicine, University of Malaya, Kuala Lumpur, Malaysia, 3 Department of Educational Psychology and Counselling, Faculty of Education, University of Malaya, Kuala Lumpur, Malaysia, 4 Department of Medicine, Faculty of Medicine, University of Malaya, Kuala Lumpur, Malaysia, 5 Department of Akidah and Islamic Thought, Academy of Islamic Studies, University of Malaya, Kuala Lumpur, Malaysia, 6 Centre for Sport and Exercise Sciences, University of Malaya, Kuala Lumpur, Malaysia, 7 Department of Music, Cultural Centre, University of Malaya, Kuala Lumpur, Malaysia, 8 Department of Nursing Science, Faculty of Medicine, University of Malaya, Kuala Lumpur, Malaysia, 9 Department of Physiology, Faculty of Medicine, University of Malaya, Kuala Lumpur, Malaysia, 10 Department of Accountancy, Faculty of Business and Accountancy, University of Malaya, Kuala Lumpur, Malaysia

* wonglp@ummc.edu.my (LPW); ivychung@ummc.edu.my (IC)

**Data Availability Statement:** All data files are available in the Kaggle database (www.kaggle.com/dataset/47a1daa9618fcb790d61e4c5cfa20363f2404c334e982d5499bf73aa8966179a).

## Abstract

Since the first nationwide movement control order was implemented on 18 March 2020 in Malaysia to contain the coronavirus disease 2019 (COVID-19) outbreak, to what extent the uncertainty and continuous containment measures have imposed psychological burdens on the population is unknown. This study aimed to measure the level of mental health of the Malaysian public approximately 2 months after the pandemic's onset. Between 12 May and 5 September 2020, an anonymous online survey was conducted. The target group included all members of the Malaysian population aged 18 years and above. The Depression Anxiety Stress Scale (DASS-21) was used to assess mental health. There were increased depressive, anxiety and stress symptoms throughout the study period, with the depression rates showing the greatest increase. During the end of the data collection period (4 August–5 September 2020), there were high percentages of reported depressive (59.2%) and anxiety (55.1%) symptoms compared with stress (30.6%) symptoms. Perceived health status was the strongest significant predictor for depressive and anxiety symptoms. Individuals with a poorer health perception had higher odds of developing depression (odds ratio [OR] = 5.68; 95% confidence interval [CI] 3.81–8.47) and anxiety (OR = 3.50; 95%CI 2.37–5.17) compared with those with a higher health perception. By demographics, young people–particularly students, females and people with poor financial conditions–were more vulnerable to mental health symptoms. These findings provide an urgent call for increased attention to detect and provide intervention strategies to combat the increasing rate of mental health problems in the ongoing COVID-19 pandemic.

**Funding:** This research was supported by Universiti Malaya COVID-19 Implementation Research Grant (RG564-2020HWB), which was granted to Ivy Chung. The funders had no role in study design, data collection and analysis, decision to publish, or preparation of the manuscript.

**Competing interests:** The authors have declared that no competing interests exist.

## Introduction

In March 2020, the World Health Organization declared coronavirus disease 2019 (COVID-19), caused by severe acute respiratory syndrome coronavirus 2 (SARS-CoV-2), a pandemic [1, 2]. COVID-19 has since caused major disruptions throughout the world, including in Malaysia. On 25 January 2020, Malaysian authorities reported the first SARS-CoV-2 infection; subsequently, on 17 March 2020, the first death was reported. Henceforth, the number of infections and deaths has increased exponentially in Malaysia. In the absence of pharmaceutical treatments and a vaccine for COVID-19, community containment measures are essential to limit the spread of SARS-CoV-2. The Malaysian government has implemented several movement control orders (MCOs) based on the current COVID-19 situation in the country. The first MCO included the closure of schools, higher education institutions and non-essential businesses (namely businesses that geared toward recreation or entertainment and those that provide services beyond the basic necessities), as well as a general prohibition of mass movements and gatherings across the country, including religious, sports, social and cultural activities. The first nationwide MCO was implemented on 18 March 2020. Since then, the country has gone through four MCO phases, all of which include the strict actions recommended by the WHO. A conditional movement control order (CMCO) was implemented from 13 May to 9 June 2020, and a recovery movement control order (RMCO) took effect from 10 June 2020 and lasted until 31 August 2020; it had more lenient restrictions. Subsequently, the RMCO was extended until 31 December 2020. To date, the Malaysian government has continuously stressed to the general public the use of face masks in public spaces, frequent hand washing and social distancing in the current ongoing pandemic. Large-scale gatherings are prohibited, but social and recreational activities and businesses are allowed to operate with social distancing measures and temperature checks in place.

Worldwide, measures to fight the COVID-19 outbreak have had tremendous social and economic impacts at both individual and country levels [3, 4]. Social distancing, self-isolation and travel restrictions have led to a reduced workforce across all economic sectors and employment loss. Closure of schools and working from home have impacted business operations, with a consequent decrease in the demand for commodities. In Malaysia, like other countries around the world, the COVID-19 pandemic has caused catastrophic economic and social disruptions [5–7]. The distancing measures, together with employment and financial insecurity, represent a massive mental health crisis affecting the wellbeing of populations throughout the world [8]. In Malaysia, the negative impact on mental health was evident during the early stages of the COVID-19 pandemic [5, 9, 10]. To date, nearly a year after the onset of the pandemic, to what extent the unpredictability and uncertainty of the COVID-19 pandemic and its prolonged physical distancing and containment measures, along with the resulting impact of the economic breakdown, has affected the mental health of the Malaysian public has never been investigated.

Therefore, the main aim of this study was to examine the level and temporal trend of mental health of the Malaysian public during the COVID-19 pandemic. In addition, factors potentially contributing to poor mental health were investigated. These findings will provide insights for the formulation of mitigation measures to help the public cope with the negative mental health effects in the currently unpredictable situation of the COVID-19 pandemic.

## Materials and methods

An anonymous Internet-based, cross-sectional survey was commenced on 12 May 2020 and ended on 5 September 2020. The inclusion criteria were that the respondents were from the general Malaysian public and $\geq$ 18 years old. The exclusion criteria were as follow: having chronic

medical conditions, pregnancy or breastfeeding, and have never had SARS-CoV-2 infection. The researchers used social network platforms (Facebook, Twitter, Instagram and WhatsApp) to disseminate and advertise the survey, entitled 'COVID-19 Health and Wellbeing Survey', to the public. Questions were presented in both English and Bahasa Malaysia in the survey link. Pilot testing was performed with 30 participants to ensure the clarity of the items and also gather suggestions for improvement. A minor revision was made based on the results of the pilot. Subsequently, the revised questionnaire was further pretested before field administration.

The first section of the survey collected demographic characteristics, participants' health status and their COVID-19 experience. Health status included participants' existing chronic diseases and self-perceived overall health status. The participants were asked to indicate whether they know of friends, neighbours or colleagues who had been diagnosed with COVID-19.

The second section assessed mental health using the Depression Anxiety Stress Scale (DASS-21) [11], which is a well-established instrument for measuring depression, anxiety and stress with good reliability and validity. Scores on three subscales–namely Depression (DASS-21-D), Anxiety (DASS-21-A) and Stress (DASS-21-S)–were generated. There are seven items in each subscale; the score of each subscale ranges from 0 to 21. The cut-offs for depression (moderate 7–10, severe 11–13 and extremely severe $\geq$ 14), anxiety (moderate 6–7, severe 8–9 and extremely severe $\geq$ 10) and stress (moderate 10–12, severe 13–16 and extremely severe $\geq$ 17) were calculated [12]. The English version of the DASS-21 has been validated for use in many Asian populations, including Malaysia [13]. The DASS-21 English version has been translated into Bahasa Malaysia and validated [14]; the Bahasa Malaysia DASS-21 has been used in many studies in Malaysia. A recent study evaluated the psychometric properties of the Bahasa Malaysia DASS-21 among non-Malays in Malaysia and revealed good reliability and validity, implying the scales can be used in a multiethnic population in Malaysia [15].

## Data analysis

Cronbach's alpha was calculated for the overall scale and the three subscales to assess reliability in terms of internal consistency. In this study, the DASS-21 had adequate to very good internal consistency, with Cronbach's alphas of 0.956 for the overall scale, 0.927 for the DASS-21-D, 0.865 for the DASS-21-A and 0.882 for the DASS-21-S.

The temporal trend of the DASS-21-D, DASS-21-A and DASS-21-S scores over the 16-week data collection period was computed. The mean and standard deviation (SD) of the DASS-21 subscale scores were divided into four equal time periods of 4-week intervals: 12 May–7 June, 8 June–5 July, 6 July–3 August and 4 August–5 September. Univariable analyses followed by multivariable logistic regression analyses, using a simultaneous forced-entry method, was used to determine the factors influencing depression, anxiety and stress. Significant predictors at $p < 0.05$ in a bivariate analysis were exported to the multivariable logistic regression model. The DASS-21-D, DASS-21-A and DASS-21-S scores were grouped into two categories: 1 = *moderate/ severe/extremely severe* and 0 = *mild/normal*. Odds ratios (ORs), 95% confidence intervals (95% CIs) and $p$ values were calculated for each independent variable. The model fit was assessed using the Hosmer–Lemeshow goodness-of-fit test [16]. Small $p$ values ($< 0.05$) mean that the model is a poor fit. All statistical analyses were performed using the Statistical Package for the Social Sciences version 20.0 (IBM Corp., Armonk, NY, USA). The level of significance was set at $p < 0.05$.

## Ethical considerations

This research was approved by the University of Malaya Research Ethics Committee (UM. TNC2/UMREC– 884). Participants were informed that their participation was voluntary, and consent was implied by completing the questionnaire.

## Results and discussion

### Demographics

A total of 1,163 complete responses were received in the 16-week data collection period. Table 1 shows the demographics of the study participants compared with the general adult population in Malaysia [17, 18]. Compared with the general Malaysian population, there was a higher percentage of female respondents, Malay ethnicity, those from the central region and those in the bottom 40% (B40) income group (< MYR4850 [USD1200] per month). The age of the participants ranged from 18 to 84 years (M = 35.2, SD = 11.9). As shown in the first and second columns in Table 2, the majority of the study participants had a diploma or were university graduates. Based on the occupation categories, nearly half were in professional and managerial occupations (50.6%), while general workers and students comprised 7.2% and 29.2%, respectively, of the participants. For all participants, 44.9% reported an average monthly household income of < MYR4000, while 31.8% reported an average monthly household income of MYR4001–8000. The majority of participants were from urban (66.1%) and suburban (26.1%) areas. The majority perceived their overall health as *very good/good* (85.9%) and

**Table 1. Comparison of demographic characteristics of the study population and the general adult population in Malaysia, 2019.**

| Characteristics | n | % Study population, n = 1163 | % Total population, n = 24510400 [*] |
|---|---|---|---|
| Age group (years) | | | |
| 18–29 | 425 | 36.5 | 38.0 |
| 30–39 | 357 | 30.7 | 22.0 |
| 40–49 | 217 | 18.7 | 15.3 |
| ≥ 50 | 164 | 14.1 | 24.7 |
| Gender | | | |
| Male | 217 | 18.7 | 51.6 |
| Female | 946 | 81.3 | 41.4 |
| Ethnicity | | | |
| Malay | 882 | 75.8 | 59.1 |
| Chinese | 100 | 8.6 | 37.4 |
| Indian | 108 | 9.3 | 29.4 |
| Others | 73 | 6.3 | 11.3 |
| Average monthly household income (MYR) (Income category group) [†] | | | |
| Below MYR4850 (B40) | 651 | 56.0 | 16.0 |
| MYR4850-10959 (M40) | 365 | 31.4 | 37.2 |
| MYR 10600 and above (T20) | 147 | 12.6 | 46.8 |
| Region[‡] | | | |
| Northern | 157 | 13.5 | 20.9 |
| Central | 697 | 59.9 | 29.7 |
| East coast | 186 | 16.0 | 13.8 |
| Southern | 80 | 6.9 | 14.5 |
| Borneo | 43 | 3.7 | 21.1 |

[*]Total number of adults 15 to 79 years of age as of 31 December 2019. Source: The 2019 Population and Housing Census of Malaysia (Census 2010) Department of Statistics Malaysia.[17]

[†]Three category of income groups: Top 20% (T20), Middle 40% (M40), and Bottom 40% (B40) in Malaysia. Source: Department of Statistics Malaysia. Household Income and Basic Amenities Survey Report 2019.[18]

[‡]Northern region (Perlis, Kedah, Perak, Penang); Central (Wilayah Persekutuan Kuala Lumpur, Selangor, Negeri Sembilan, Putrajaya); East coast (Terengganu, Kelantan, Pahang); Southern (Melaka, Johor); Borneo (Sabah, Sarawak, Labuan)

**Table 2. Univariable and multivariable analyses of factors associated with symptoms of depression, anxiety and stress (N = 1163).**

| Socio demographic characteristics | Overall N(%) | Depression Moderate/ Severe/ Extremely severe (n = 344) | p-value | Adjusted OR (95% CI)[a] | Anxiety Moderate/ Severe/ Extremely severe (n = 461) | p-value | Adjusted OR (95% CI)[b] | Stress Moderate/ Severe/ Extremely severe (n = 202) | p-value | Adjusted OR (95% CI)[c] |
|---|---|---|---|---|---|---|---|---|---|---|
| **Age group (years)** | | | | | | | | | | |
| 18–25 | 322 (27.7) | 156 (48.4) | | 2.03 (1.06–3.91)* | 182 (56.5) | | 1.35 (0.76–2.39) | 96 (29.8) | | 2.73 (1.19–6.23)* |
| 26–45 | 592 (50.9) | 155 (26.2) | p<0.001 | 2.07 (1.30–3.30)** | 219 (37.0) | p<0.001 | 1.43 (0.99–2.08) | 93 (15.7) | p<0.001 | 3.23 (1.72–6.31)*** |
| > 45 | 249 (21.4) | 33 (13.3) | | Ref | 60 (24.1) | | Ref | 13 (5.2) | | Ref |
| **Gender** | | | | | | | | | | |
| Male | 217 (18.7) | 56 (25.8) | 0.188 | | 73 (33.6) | 0.046 | Ref | 27 (12.4) | 0.037 | Ref |
| Female | 946 (81.3) | 288 (30.4) | | | 388 (41.0) | | 1.49 (1.06–2.10)* | 175 (18.5) | | 1.83 (1.14–2.95)* |
| **Ethnicity** | | | | | | | | | | |
| Malay | 882 (75.8) | 254 (28.8) | | | 362 (41.0) | | 1.09 (0.64–1.86) | 154 (17.5) | | |
| Chinese | 100 (8.6) | 34 (34.0) | 0.320 | | 39 (39.0) | 0.039 | 0.69 (0.35–1.36) | 22 (22.0) | 0.202 | |
| Indian | 108 (9.3) | 29 (26.9) | | | 29 (26.9) | | 0.50 (0.25–0.99)* | 12 (11.1) | | |
| Bumiputera Sabah/ Sarawak/ Others | 73 (6.3) | 27 (37.0) | | | 31 (42.5) | | Ref | 14 (19.2) | | |
| **Marital Status** | | | | | | | | | | |
| Never married | 490 (42.1) | 223 (45.5) | p<0.001 | 1.71 (0.92–3.16) | 262 (53.5) | p<0.001 | 1.40 (0.82–2.40) | 136 (27.8) | p<0.001 | 2.05 (0.94–4.46) |
| Ever married | 673 (57.9) | 121 (18.0) | | Ref | 199 (29.6) | | Ref | 66 (9.8) | | Ref |
| **Have child/ children** | | | | | | | | | | |
| Yes | 595 (51.2) | 104 (17.5) | p<0.001 | Ref | 172 (28.9) | p<0.001 | Ref | 58 (9.7) | p<0.001 | Ref |
| No | 568 (48.8) | 240 (42.3) | | 1.17 (0.65–2.09) | 289 (50.9) | | 1.17 (0.71–1.92) | 144 (25.4) | | 0.93 (0.44–1.96) |
| **Highest educational level** | | | | | | | | | | |
| Secondary and below | 78 (6.7) | 12 (15.4) | 0.004 | Ref | 18 (23.1) | 0.002 | Ref | 6 (7.7) | 0.019 | Ref |
| Tertiary | 1085 (93.3) | 332 (30.6) | | 1.82 (0.89–3.71) | 443 (40.8) | | 1.80 (0.99–3.250 | 196 (18.1) | | 1.85 (0.72–4.71) |
| **Occupation type** | | | | | | | | | | |
| Professional and managerial | 589 (50.6) | 116 (19.7) | | Ref | 187 (31.7) | | Ref | 66 (11.2) | | Ref |
| General worker | 84 (7.2) | 16 (19.0) | p<0.001 | 0.85 (0.45–1.62) | 21 (25.0) | p<0.001 | 0.71 (0.41–1.23) | 4 (4.8) | p<0.001 | 0.39 (0.13–1.14) |
| Housewife/ Retired/ Other | 150 (12.9) | 38 (25.3) | | 1.20 (0.73–1.96) | 51 (34.0) | | 1.04 (0.68–1.60) | 21 (14.0) | | 1.39 (0.77–2.52) |
| Student | 340 (29.2) | 174 (51.2) | | 1.79 (1.12–2.86)* | 202 (59.4) | | 1.84 (1.19–2.85)** | 111 (32.6) | | 2.03 (1.18–3.56)* |

*(Continued)*

**Table 2.** (Continued)

| Socio demographic characteristics | Overall N(%) | Depression Moderate/ Severe/ Extremely severe (n = 344) | p-value | Adjusted OR (95% CI)[a] | Anxiety Moderate/ Severe/ Extremely severe (n = 461) | p-value | Adjusted OR (95% CI)[b] | Stress Moderate/ Severe/ Extremely severe (n = 202) | p-value | Adjusted OR (95% CI)[c] |
|---|---|---|---|---|---|---|---|---|---|---|
| Monthly average household income (MYR) | | | | | | | | | | |
| 4000 and below | 522 (44.9) | 199 (38.1) | | 0.90 (0.59–1.39) | 244 (46.7) | | 1.03 (0.70–1.52) | 113 (21.6) | | 0.49 (0.30–0.80)** |
| 4001–8000 | 370 (31.8) | 76 (20.5) | p<0.001 | 0.81 (0.53–1.24) | 120 (32.4) | p<0.001 | 0.92 (0.64–1.32) | 36 (9.7) | p<0.001 | 0.43 (0.26–0.71)** |
| >8000 | 271 (23.3) | 69 (25.5) | | Ref | 97 (35.8) | | Ref | 53 (19.6) | | Ref |
| Perceived current financial status | | | | | | | | | | |
| Poor | 125 (10.7) | 71 (56.8) | | 2.63 (1.51–4.59)** | 67 (53.6) | | 1.28 (0.76–2.14) | 40 (32.0) | | 1.93 (1.03–3.61)* |
| Medium | 732 (62.9) | 211 (28.8) | p<0.001 | 1.36 (0.93–1.98) | 296 (40.4) | p<0.001 | 1.27 (0.92–1.76) | 124 (16.9) | p<0.001 | 1.46 (0.93–2.29) |
| Good | 306 (26.3) | 62 (20.3) | | Ref | 98 (32.0) | | Ref | 38 (12.4) | | Ref |
| Locality | | | | | | | | | | |
| Urban | 723 (62.2) | 212 (29.3) | | | 268 (37.1) | | | 126 (17.4) | | |
| Sub-urban | 304 (26.1) | 91 (29.9) | 0.969 | | 137 (45.1) | 0.053 | | 54 (17.8) | 0.919 | |
| Rural | 136 (11.7) | 41 (30.1) | | | 56 (41.2) | | | 22 (16.2) | | |
| Region | | | | | | | | | | |
| Northern | 157 (13.5) | 56 (35.7) | | 0.70 (0.36–1.33) | 61 (38.9) | | | 33 (21.0) | | |
| Southern | 80 (6.9) | 23 (28.8) | | 0.95 (0.62–1.45) | 34 (42.5) | | | 11 (13.8) | | |
| Central | 697 (59.9) | 214 (30.7) | 0.042 | 0.66 (0.38–1.14) | 278 (39.9) | 0.877 | | 127 (18.2) | 0.212 | |
| East Coast | 186 (16.0) | 39 (21.0) | | 1.12 (0.49–2.55) | 69 (37.1) | | | 23 (12.4) | | |
| Borneo Island | 43 (3.7) | 12 (27.9) | | Ref | 19 (44.2) | | | 8 (18.6) | | |
| Health status | | | | | | | | | | |
| Diagnosed with any chronic diseases | | | | | | | | | | |
| Yes | 99 (8.5) | 35 (35.4) | 0.205 | | 45 (45.5) | 0.238 | | 21 (21.2) | 0.331 | |
| No | 1064 (91.5) | 309 (29.0) | | | 416 (39.1) | | | 181 (17.0) | | |
| Perceived health status | | | | | | | | | | |
| Very good/ Good | 999 (85.9) | 233 (23.3) | p<0.001 | Ref | 351 (35.1) | p<0.001 | Ref | 133 (13.3) | p<0.001 | Ref |
| Fair/ Poor/ Very poor | 164 (14.1) | 111 (67.7) | | 5.68 (3.81–8.47)*** | 110 (67.1) | | 3.50 (2.37–5.17)*** | 69 (42.1) | | 3.66 (2.46–5.45)*** |
| COVID-19 experience | | | | | | | | | | |

*(Continued)*

**Table 2.** (Continued)

| Socio demographic characteristics | Overall N(%) | Depression | | | Anxiety | | | Stress | | |
| | | Moderate/ Severe/ Extremely severe (n = 344) | p-value | Adjusted OR (95% CI)[a] | Moderate/ Severe/ Extremely severe (n = 461) | p-value | Adjusted OR (95% CI)[b] | Moderate/ Severe/ Extremely severe (n = 202) | p-value | Adjusted OR (95% CI)[c] |
|---|---|---|---|---|---|---|---|---|---|---|
| Ever known anyone infected or died of COVID-19 | | | | | | | | | | |
| Yes | 272 (23.4) | 70 (25.7) | 0.129 | | 114 (41.9) | 0.396 | | 38 (14.0) | 0.100 | |
| No | 891 (76.6) | 274 (30.8) | | | 347 (38.9) | | | 164 (18.4) | | |

[*],p<0.05

[**]p<0.01

[***]p<0.001

[a]Hosmer–Lemeshow test, chi-square: 8.048, p-value: 0.429; Nagelkerke $R^2$: 0.280

[b]Hosmer–Lemeshow test, chi-square: 9.609, p-value: 0.294; Nagelkerke $R^2$: 0.180

[c]Hosmer–Lemeshow test, chi-square: 6.584, p-value: 0.582; Nagelkerke $R^2$: 0.229

the majority (91.5%) did not have any chronic diseases. Only 23.4% reported knowing someone (family members, friends, neighbours or colleagues) who had been diagnosed with or died from COVID-19.

Fig 1 shows the trend of confirmed COVID-19 cases in Malaysia and the recruitment period. Fig 2 shows the temporal trend of the average DASS-21-D, DASS-21-A and DASS-21-S subscale scores across the four successive time periods in the 16-week data collection period. The average DASS-21-S score (M = 10.1, SD = 7.6) during the first four weeks of data collection was higher than the average DASS-21-D (M = 7.5, SD = 7.7) and DASS-21-A (M = 7.0, SD = 6.7) scores. The average DASS-21-D score was markedly elevated across the data collection period. The average DASS-21-D score (M = 17.9, SD = 11.8) was highest in the last four weeks of the data collection period, followed by the DASS-21-S (M = 15.6, SD = 9.6) and DASS-21-A (M = 11.9, SD = 10.0) scores. Fig 3 shows the percentage of participants with depressive, anxiety and stress symptoms according to the DASS-21 scores for each time period. The highest percentage of respondents reported anxiety (32.3%), followed by depressive (20.6%) and stress (11.5%) symptoms in the first time period (12 May–7 June 2020). In the

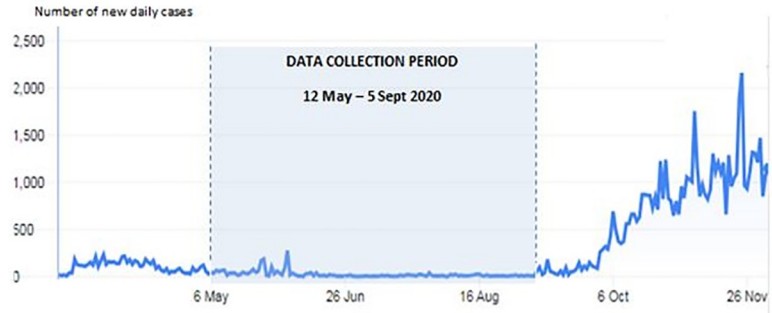

**Fig 1. Data collection period and the trend of confirmed cases of COVID-19 in Malaysia.**

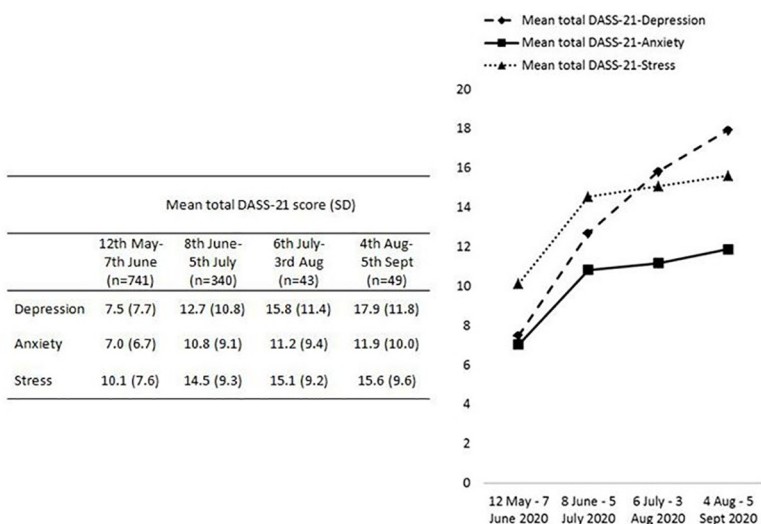

**Fig 2. Temporal changes in mean total DASS-21 score at different time point.**

fourth time period (4 August–5 September 2020), a higher percentage reported depressive (59.2%) and anxiety (55.1%) symptoms compared with stress (30.6%) symptoms.

Table 2 shows the percentage of participants with depressive, anxiety and stress symptoms and their associated factors during the 16-week study period. On average, depressive, anxiety and stress symptoms were reported in 21.3% (n = 344), 28.6% (n = 461) and 12.5% (n = 202), respectively, of the participants during the study period. Multivariable logistic regression analysis showed that the 18–25-year-old and 26–45-year-old groups had twice the odds for depressive symptoms than the > 45-year-old group. Students and participants who perceived a poor

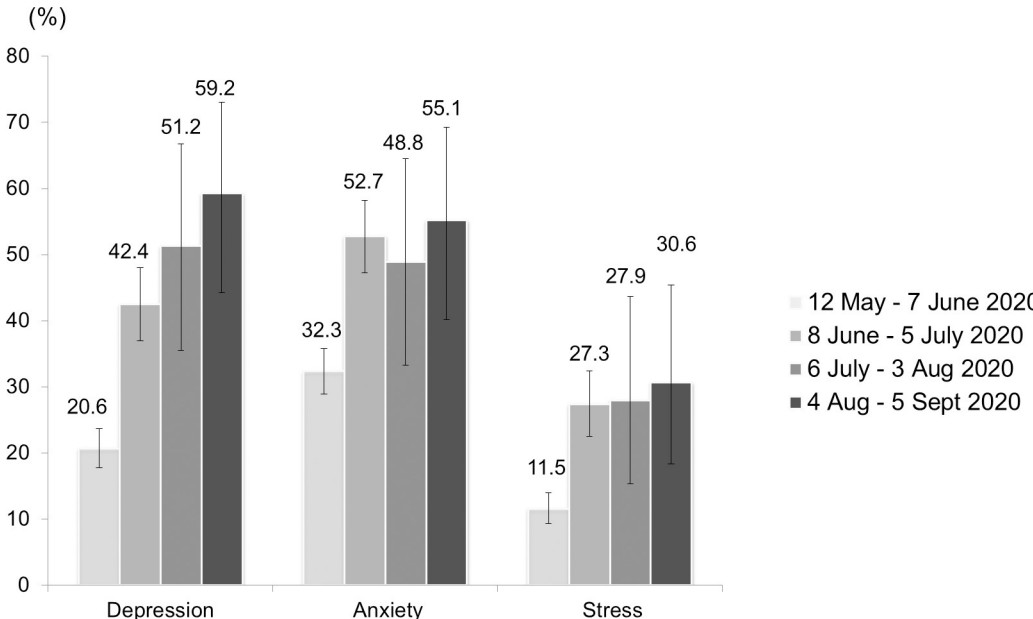

**Fig 3. Proportion of participants with the presence of depressive, anxiety and stress symptoms by the four time point of the study period.**

financial status reported a significantly higher likelihood of depressive symptoms. Females reported a significantly higher likelihood of having anxiety disorder than males (OR = 1.49, 95% CI 1.06–2.10). Students had higher odds of anxiety symptoms than the professional and managerial occupational groups (OR = 1.84, 95% CI 1.19–2.85).

In the multivariable regression analysis of factors influencing stress symptoms, the 25–45-year-old group (OR = 3.23, 95% CI 1.72–6.31) and the 18–25-year-old group (OR = 2.73, 95% CI 1.19–6.23) showed higher odds of stress symptoms than the > 45-year-old group. Those who perceived a poor financial status (OR = 1.93, 95% CI 1.03–3.61) had nearly 2 times higher odds of stress symptoms than those who perceived a good financial status. Although a higher percentage of participants with a monthly income < MYR4000 reported stress symptoms, in the multivariable logistic regression analysis, the highest income group had higher odds of stress symptoms. Perceived health status was the strongest significant predictor for depressive and anxiety symptoms. The odds of depression (OR = 5.68, 95% CI 3.81–8.47) and anxiety (OR = 3.50, 95% CI 2.37–5.17) were markedly higher in respondents who perceived a poorer health status.

This study examined the depressive, anxiety and stress symptoms of the Malaysian public during the implementation of a CMCO and RMCO and explored its associated factors using a cross-sectional study design. The data collection period was between the second and third waves of the COVID-19 pandemic in Malaysia, and the country is still facing the continuous threat of the disease.

In this study, responses were obtained during the period when the government imposed more lenient containment measures, and the results revealed increasing levels of depressive, anxiety and stress symptoms throughout the study duration. In line with previous studies from Hong Kong [19] and the United Kingdom [20], the public seemed to experience increased symptoms of mental illness as the pandemic progressed over time. Although psychological well-being of the public is expected to improve when restrictions were lifted, nonetheless, the negative mental impact of the people in this study did not decline despite the shift from CMCO to RMCO. The psychological impact continues to rise across the CMCO and RMCO phase. The increase in mental disorders over time can perhaps be seen as part of the continuous economic and societal consequences as the pandemic has progressed. Many countries, including Malaysia, continue to adopt unprecedented physical distancing policies. As the banning of mass gatherings, work-from-home policies and virtual meetings have been put into place during the COVID-19 pandemic, many industries have been affected negatively. Many individuals continue to suffer mentally as economic consequences continue, in addition to restrictions on social activities and prolonged confinement to their homes.

It is worth highlighting the marked increase in depressive symptoms compared with anxiety and stress symptoms. This finding implies that the Malaysian public became increasingly vulnerable to depressive disorder as the pandemic continued. Furthermore, during the last four weeks of data collection, when the COVID-19 pandemic moved into its seventh month in Malaysia, the percentage of participants with depressive symptoms was highest: the prevalence of depression among the study participants was close to 60% based on the DASS-21-D score. Of note, depression is a leading cause of disability around the world and contributes greatly to the global burden of disease [21]. In addition, the percentage of anxiety (55.1%) and stress (30.6%) symptoms were highest at the end of the data collection period. These findings further indicate the seriousness of overall mental health problems as the COVID-19 pandemic has progressed. There is evidence that mental disorders are associated with suicidal ideations and attempts [22]. Specifically, depression was found to be the main risk factors associated with suicidal behaviours [23, 24]. Malaysia has seen a rise in suicide cases and attempts during the COVID-19 pandemic [25, 26]. These findings warrant urgent monitoring of the Malaysian

population's mental health and prompt provision of counselling to mitigate the detrimental impact on society.

In this study, the 18–25-year-old group, followed by the 26–45-year-old group, reported more anxiety, depressive and stress symptoms. Most of the 18–25-year-old respondents were college or university students. Several population-based surveys have also revealed that students report a higher prevalence of depressive, anxiety and stress symptoms [27, 28]. Studies on student populations throughout the world [29–31], including in Malaysia [32], have shown that the COVID-19 pandemic has put a strain on their mental health. The main reasons for students experiencing heightened psychological distress include economic effects, changes in academic activities, difficulties adapting to online distance learning methods and uncertainty about the future with regard to academics and careers. Given the unprecedented COVID-19 crisis and uncertainly with regard to how long schools will remain closed, the findings suggest the importance of closely monitoring students' mental health status and providing psychological counseling or mental health services. Schools and health authorities should work together to deliver prompt psychological support to affected students.

This study found a higher prevalence of depressive, anxiety, stress symptoms in women compared with men. Specifically, the multivariable analyses revealed a significant increased risk of anxiety and stress in females compared with males. These findings are in line with previous studies from around the world [33–36] as well as Malaysia [5], where women seem to have experienced elevated psychological symptoms related to the COVID-19 pandemic compared with men. The sex difference is in line with evidence suggesting that women are generally more vulnerable to stress and anxiety disorders in response to traumatic events, while men show better resilience [37, 38]. It has been suggested that intervention strategies and policies for public health emergencies should consider a gender-specific approach to address mental health inequities [39].

Financial distress due to the pandemic has been reported as a key correlate of poor mental health [40]. In this study, univariable analyses revealed that higher income and perceived financial status were well associated with better mental health outcomes. However, in the multivariable analyses, only perceived financial status was a significant predictor of better mental health outcomes. This could possibly be explained because being a high-income earner does not necessarily imply a strong financial situation. By contrast, people with strong financial situations–for example, with adequate savings–showed stronger resilience to the financial crisis during the COVID-19 pandemic. The findings indicate that minimising financial disruption during the COVID-19 pandemic should be a central goal of public health policy.

In this study, people with a fair, poor or very poor health status were more likely to report negative mental health impacts compared with those with a very good or good health status. In addition, health status was the strongest predictor for depressive and anxiety symptoms in the multivariable regression findings. These findings are in line with a recent study reporting that individuals with more medical comorbidities were more likely to report elevated depressive symptoms during the COVID-19 pandemic [41]. A study found that reduced access to routine medical care during the pandemic resulted in greater mental health among people with medical comorbidities [41]. This could be the indicator that psychological strain of pre-existing medical comorbidities may have been further increased during the COVID-19 pandemic. Other unprecedented life situations during the pandemic, including job loss, financial hardship and sedentary behaviours, may also worsen preexisting comorbidities and thereby exacerbate mental health problems. The consequences of the COVID-19 pandemic on the mental health of people with pre-existing health conditions warrant ensuring patients' are sufficiently self-aware to realise when they need to seek help. In addition, family members and providers should be encouraged to support individuals to promote help-seeking behaviour. People with

pre-existing comorbidities would need extra support to cultivate resilience, coping, mindfulness and healthy adjustment with the change to an inactive, sedentary lifestyle to enhance their mental wellbeing during the pandemic.

Some limitations of our study should be noted. The first limitation is the cross-sectional design: we identified associations but could not infer cause and effect. Another potential issue is the influence of selection bias on the prevalence of mental health problems seen in this sample. We were careful to ensure that the recruitment advertisement did not mention the topic of mental health; we advertised the study as a 'COVID-19 Health and Wellbeing Survey'. Nonetheless, it is possible that people experiencing psychological distress or mental health consequences were more likely to respond to the survey. It is also important to note that online survey methods may lead to a biased response towards people who have good Internet literacy and access as well as those who are more educated. This bias may have a disproportionate impact on subsections of the population, such as high percentage of participants in this study who have a diploma or are university graduates. The study also has a higher representation of female participants. Nonetheless, we did obtain a sample that was relatively representative of the Malaysian population based on age and location. It is also important to note that our study used both the English and the Bahasa Malaysia version of DASS-21 in the same questionnaire. To our best knowledge, the measurement of invariance across the English and Bahasa Malaysia version of DASS-21 has never been examined before. Unfortunately, the measurement invariance across the English and Bahasa Malaysia version of DASS-21 was unable to be determined in our bilingual survey link. In the absence of measurement invariance procedures across the English and Bahasa Malaysia version of DASS-21 in this study, the psychometric robustness associated with different interpretations of items in the two languages was unknown. Testing for and assuring measurement invariance across different languages, culture or group comparisons is essential [42–44]. Future studies on DASS-21 should include validation across different group comparisons and testing of invariance of all versions. Lastly, the study is also limited by the small number of responses in the last two time periods, hence results should be interpreted with caution. Despite these limitations, our results are in line with many previous studies.

## Conclusion

Our study shows that mental health symptoms, especially depression and anxiety, have been overwhelmingly prevalent in the Malaysian population as the COVID-19 pandemic has progressed. Although the number of COVID-19 cases during the data collection period was relatively low, there was a continuous increase in the percentage of respondents with depressive, anxiety and stress symptoms, implying a cumulative mental health burden. The increase in the rate of depression was most rapid, and the rate of depression was dramatically high as the COVID-19 pandemic moved into its seventh month in Malaysia. Poor health status was the strongest significant predictor for depressive and anxiety symptoms. By demographics, young people, particularly students, females and people with poor financial conditions, were more vulnerable to mental health symptoms. These findings emphasise the necessity of monitoring mental health disorders among the public during the COVID-19 pandemic and subsequently providing targeted interventions for people at elevated risks, as identified in this study.

## Acknowledgments

We thank the assistance provided by all members in the Caring Together team.

## Author Contributions

**Conceptualization:** Li Ping Wong, Azmawaty Mohamad Nor, Maw Pin Tan, Diana Lea Baranovich, Che Zarrina Saari, Sareena Hanim Hamzah, Ku Wing Cheong, Chiew Hwa Poon, Vimala Ramoo, Chong Chin Che, Kyaimon Myint, Suria Zainuddin, Ivy Chung.

**Data curation:** Li Ping Wong, Haridah Alias.

**Formal analysis:** Li Ping Wong, Haridah Alias.

**Funding acquisition:** Ivy Chung.

**Investigation:** Li Ping Wong, Afiqah Alyaa Md Fuzi, Intan Sofia Omar, Azmawaty Mohamad Nor, Maw Pin Tan, Diana Lea Baranovich, Che Zarrina Saari, Sareena Hanim Hamzah, Ku Wing Cheong, Chiew Hwa Poon, Vimala Ramoo, Chong Chin Che, Kyaimon Myint, Suria Zainuddin, Ivy Chung.

**Methodology:** Li Ping Wong, Haridah Alias.

**Project administration:** Afiqah Alyaa Md Fuzi, Intan Sofia Omar.

**Resources:** Ivy Chung.

**Visualization:** Li Ping Wong, Haridah Alias.

**Writing – original draft:** Li Ping Wong.

**Writing – review & editing:** Li Ping Wong, Azmawaty Mohamad Nor, Maw Pin Tan, Diana Lea Baranovich, Che Zarrina Saari, Sareena Hanim Hamzah, Ku Wing Cheong, Chiew Hwa Poon, Vimala Ramoo, Chong Chin Che, Kyaimon Myint, Suria Zainuddin, Ivy Chung.

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
