## [Decision Letter · Decision Letter 0]

26 Feb 2021

PONE-D-21-02085

Escalating progression of mental health disorders during the COVID-19 pandemic: evidence from a nationwide survey

PLOS ONE

Dear Dr. Wong,

Thank you for submitting your manuscript to PLOS ONE. After careful consideration, we feel that it has merit but does not fully meet PLOS ONE’s publication criteria as it currently stands. Therefore, we invite you to submit a revised version of the manuscript that addresses the points raised during the review process.

One expert and I have reviewed your submission and both of us agree that your submission has some merits. However, there are some serious concerns raised by the reviewer and you have to carefully consider them and address them in your revision. Among these good comments made by the reviewer, I would like to highlight the problem of using two versions of DASS-21 in the present study. The authors should have justified why the two language versions can be combined using in the present study. If the authors failed to provide a good justification, I am afraid that it will be hard to accept this paper.

We look forward to receiving your revised manuscript.

Kind regards,

Chung-Ying Lin

Academic Editor

PLOS ONE

Journal Requirements:

Reviewers' comments:

Reviewer's Responses to Questions

**Comments to the Author**

1. Is the manuscript technically sound, and do the data support the conclusions?

Reviewer #1: Yes

2. Has the statistical analysis been performed appropriately and rigorously? 

Reviewer #1: Yes

3. Have the authors made all data underlying the findings in their manuscript fully available?

Reviewer #1: Yes

4. Is the manuscript presented in an intelligible fashion and written in standard English?

Reviewer #1: Yes

5. Review Comments to the Author

Reviewer #1: General comment:

The manuscript entitled “Escalating progression of mental health disorders during the COVID-19 pandemic: evidence from a nationwide survey” highlighted the mental health concerns of the Malaysian population in the COVID-19 pandemic. The strength of the manuscript is the timely assessment during COVID-19 with large sample size. However, some parts of the manuscript should be elucidated and clarified.

Mainly, different MCOs was introduced, yet the purpose of this introduction was not clear. Furthermore, the rationale of dividing four time periods was not explained well. Is it related to the MCOs, or due to other reasons? Please see the specific comments for details.

Specific comments:

Introduction

(1) For the first paragraph, “The first MCO included the closure of schools, higher education institutions and ‘non-essential’ businesses”.

Why quotation mark was used? Is there any special meaning for the non-essential? Authors may provide examples for the non-essential businesses.

(2) Authors introduced different MCOs in the first paragraph. Is the author simply tried to illustrate the situation in Malaysia, or tried to illustrate the impact of COVID-19 and related MCOs?

If authors think MCOs is important in this study, they should discuss the results incorporate with MCOs. For example, authors mentioned RMCO had more lenient restrictions compared to CMCO. Should we expect the negative impact on people’s daily living was reduced under RMCO, and hence, may influence mental health?

Materials and methods

(1) Any exclusion criteria for the recruitment?

(2) “The participants were asked to indicate whether they know of friends, neighbours or colleagues who had been diagnosed with COVID-19.”

How about the participants themselves? Whether participants had been diagnosed with COVID-19 might be important to their mental health.

(3) In this study, both English version and Bahasa Malaysia version of DASS-21 were used and pooled for analysis. Could authors provide reference to support the measurement invariance across these two language versions?

Data analysis

(1) I suggest to report internal consistency for English version and Bahasa Malaysia version separately.

(2) Could authors explain the rationale of dividing four time periods? Without any explanation, it made me wonder whether it was related to the MCOs. However, it seems that they did not match.

Results and discussion

(1) As the study divided four time periods, the number of participants in each period should be provided.

(2) For those who reported their gender, most of them were female. Authors may consider to discuss it in the limitation.

6. PLOS authors have the option to publish the peer review history of their article (what does this mean?). If published, this will include your full peer review and any attached files.

Reviewer #1: No

---

## [Author Response · Author response to Decision Letter 0]

2 Mar 2021

Reply: We do not have ethical or legal restrictions on sharing a de-identified data set, we noted in the cover letter. We have also uploaded our data in data repository Havard dataverse (https://doi.org/10.7910/DVN/COITOK)

Reply: The data is uploaded in the following data repository Havard dataverse

Link database: https://doi.org/10.7910/DVN/COITOK

 Reply: Added

Reviewers' comments:

Reviewer's Responses to Questions

Comments to the Author

1. Is the manuscript technically sound, and do the data support the conclusions?

Reviewer #1: Yes

2. Has the statistical analysis been performed appropriately and rigorously?

Reviewer #1: Yes

3. Have the authors made all data underlying the findings in their manuscript fully available?

 Data has been deposited in data repository (https://doi.org/10.7910/DVN/COITOK)

Reviewer #1: Yes

4. Is the manuscript presented in an intelligible fashion and written in standard English?

Reviewer #1: Yes

5. Review Comments to the Author

Reviewer #1: General comment:

The manuscript entitled “Escalating progression of mental health disorders during the COVID-19 pandemic: evidence from a nationwide survey” highlighted the mental health concerns of the Malaysian population in the COVID-19 pandemic. The strength of the manuscript is the timely assessment during COVID-19 with large sample size. However, some parts of the manuscript should be elucidated and clarified.

Mainly, different MCOs was introduced, yet the purpose of this introduction was not clear. Furthermore, the rationale of dividing four time periods was not explained well. Is it related to the MCOs, or due to other reasons? Please see the specific comments for details.

Specific comments:

Introduction

(1) For the first paragraph, “The first MCO included the closure of schools, higher education institutions and ‘non-essential’ businesses”.

Why quotation mark was used? Is there any special meaning for the non-essential? Authors may provide examples for the non-essential businesses.

Reply: We remove ‘ ‘

Added: meaning of non-essential businesses, line 72-74

The first MCO included the closure of schools, higher education institutions and non-essential businesses (namely businesses that geared toward recreation or entertainment and those that provide services beyond the basic necessities),……..

(2) Authors introduced different MCOs in the first paragraph. Is the author simply tried to illustrate the situation in Malaysia, or tried to illustrate the impact of COVID-19 and related MCOs?

If authors think MCOs is important in this study, they should discuss the results incorporate with MCOs. For example, authors mentioned RMCO had more lenient restrictions compared to CMCO. Should we expect the negative impact on people’s daily living was reduced under RMCO, and hence, may influence mental health?

Reply: Thank you for highlighting the shortcoming. In line 135-6, we noted “The temporal trend of the DASS-21-D, DASS-21-A and DASS-21-S scores over the 16-week data collection period was computed”. Hence we added in line 98 the objective of investigating the temporal trend of mental health.

Therefore, the main aim of this study was to examine the level and temporal trend of mental health of the Malaysian public during the COVID-19 pandemic.

In Introduction, we provide a snapshots of the different phases of MCOs in Malaysia, however, the data collection was conducted after during CMCO and RMCO, where the government imposed more lenient containment measures.

Added, although ideally when restrictions were lifted, mental health improve, but out study found otherwise. The possible reasons were noted in line 245-252.

Added line 242-245

Although psychological well-being of the public is expected to improve when restrictions were lifted, nonetheless, the negative mental impact of the people in this study did not decline despite the shift from CMCO to RMCO. The psychological impact continues to rise across the CMCO and RMCO phase.

Materials and methods

(1) Any exclusion criteria for the recruitment?

Reply: Added exclusion criteria, line 108

The exclusion criteria were as follow: having chronic medical conditions, pregnancy or breastfeeding, and have never had SARS-CoV-2 infection.

(2) “The participants were asked to indicate whether they know of friends, neighbours or colleagues who had been diagnosed with COVID-19.”

How about the participants themselves? Whether participants had been diagnosed with COVID-19 might be important to their mental health.

Reply: The study did not include people who have had been diagnosed with COVID-19 as noted in the added exclusion criteria.

(3) In this study, both English version and Bahasa Malaysia version of DASS-21 were used and pooled for analysis. Could authors provide reference to support the measurement invariance across these two language versions?

Reply: To our best knowledge, to date, there is no reference to support the measurement invariance across the English and Bahasa Malaysia version of the DASS-21. Unfortunately, in our study, the English and Bahasa Malaysia questions of the DASS-21 were place side by side and hence the measurement of invariance across the English and BM versions of the DASS-21 is unable to be calculated in the current study. 

We reported the limitation of the measurement invariance across these two language versions.

Added line 329

It is also important to note that this study used both the English and the Bahasa Malaysia version of FASS-21, however, the measurement invariance across the English and Bahasa Malaysia version of DASS-21 was unable to be both version of the DASS-21 were included in the same survey link.

Data analysis

(1) I suggest to report internal consistency for English version and Bahasa Malaysia version separately.

Reply: Thank you for suggestion, unfortunately in our survey, both Bahasa Malaysia and English questions were incorporated into our sole survey link. Each questions is bilingual, hence we are unable to report internal consistency for both version. We are unsure if the respondent were reading the English or BM version when answering the questions. 

(2) Could authors explain the rationale of dividing four time periods? Without any explanation, it made me wonder whether it was related to the MCOs. However, it seems that they did not match.

Reply: The four time periods were divided by equal time intervals. We introduces the phases of MCO, CMCO and RMCO in the Introduction to provide information of all stages of movement control in Malaysia during the pandemic, however, our data collection was conducted during CMCO and RMCO. Hence, rational of dividing to four time periods was not the sequence of the MCO, CMCO and RMCO. We could only relate the findings on the two phases of MCOs. 

Added “by equal intervals”, hence added the word “equal” line line 138

The mean and standard deviation (SD) of the DASS-21 subscale scores were divided into four equal time periods of 4-week intervals….

Results and discussion

(1) As the study divided four time periods, the number of participants in each period should be provided.

Reply: We noted the number of participants in each phases in Fig. 2. We made minor typo mistakes on the number of responses, corrected them in text and tables, and shown in the correction tracking. The rest of the information is accurate.

We added in the limitation of the low number of responses in the last two time periods. Line 332.

Lastly, the study is also limited by the small number of responses in the last two time periods, hence results should be interpreted with caution. 

(2) For those who reported their gender, most of them were female. Authors may consider to discuss it in the limitation.

Added in line 327.

The study also has a higher representation of female participants.

---

## [Editor Report · Decision Letter 1]

4 Mar 2021

PONE-D-21-02085R1

Escalating progression of mental health disorders during the COVID-19 pandemic: evidence from a nationwide survey

PLOS ONE

Dear Dr. Wong,

Thank you for submitting your manuscript to PLOS ONE. After careful consideration, we feel that it has merit but does not fully meet PLOS ONE’s publication criteria as it currently stands. Therefore, we invite you to submit a revised version of the manuscript that addresses the points raised during the review process.

In general, the responses are decent and I only have one concern regarding the measurement invariance.

Apparently, the authors cannot justify why the two language versions can be combined used and thus they put this into their limitation. I think that putting this as one of the limitations is appropriate. However, the authors should emphasize this issue and encourage future studies to investigate the measurement invariance. The authors should also mention why it is important to investigate measurement invariance with proper citation. They can take reference from Leung et al. (2020).

Leung, H., Pakpour, A. H., Strong, C., Lin, Y. C., Tsai, M. C., Griffiths, M. D., Lin, C. Y., & Chen, I. H. (2020). Measurement invariance across young adults from Hong Kong and Taiwan among three internet-related addiction scales: Bergen Social Media Addiction Scale (BSMAS), Smartphone Application-Based Addiction Scale (SABAS), and Internet Gaming Disorder Scale-Short Form (IGDS-SF9) (Study Part A). Addictive behaviors, 101, 105969. https://doi.org/10.1016/j.addbeh.2019.04.027

We look forward to receiving your revised manuscript.

Kind regards,

Chung-Ying Lin

Academic Editor

PLOS ONE
---

## [Author Response · Author response to Decision Letter 1]

5 Mar 2021

Reply: We emphasized the importance of the issue and the importance of future studies to investigate the measurement invariance.

Testing for and assuring measurement invariance across different languages, culture or group comparisons is essential [42-44]. Future studies on DASS-21 should include validation across different group comparisons and testing of invariance of all versions.

Reply: We noted that the importance of investigating measurement invariance to ensure psychometric robustness.

In the absence of measurement invariance procedures across the English and Bahasa Malaysia version of DASS-21 in this study, the psychometric robustness associated with different interpretations of items in the two languages was unknown.

Reply: We rephrased the limitation section as follow.

To our best knowledge, the measurement of invariance across the English and Bahasa Malaysia version of DASS-21 has never been examined before. Unfortunately, the measurement invariance across the English and Bahasa Malaysia version of DASS-21 was unable to be determined in our bilingual survey link. In the absence of measurement invariance procedures across the English and Bahasa Malaysia version of DASS-21 in this study, the psychometric robustness associated with different interpretations of items in the two languages was unknown. Testing for and assuring measurement invariance across different languages, culture or group comparisons is essential [42-44]. Future studies on DASS-21 should include validation across different group comparisons and testing of invariance of all versions.

---

## [Editor Report · Decision Letter 2]

9 Mar 2021

Escalating progression of mental health disorders during the COVID-19 pandemic: evidence from a nationwide survey

PONE-D-21-02085R2

Dear Dr. Wong,

We’re pleased to inform you that your manuscript has been judged scientifically suitable for publication and will be formally accepted for publication once it meets all outstanding technical requirements.

Kind regards,

Chung-Ying Lin

Academic Editor

PLOS ONE
---

## [Editor Report · Acceptance letter]

15 Mar 2021

PONE-D-21-02085R2 

Escalating progression of mental health disorders during the COVID-19 pandemic: evidence from a nationwide survey 

Dear Dr. Wong:

I'm pleased to inform you that your manuscript has been deemed suitable for publication in PLOS ONE. Congratulations! Your manuscript is now with our production department. 

Kind regards, 

on behalf of

Dr. Chung-Ying Lin 

Academic Editor

PLOS ONE